# Effect of Ridge Height, Row Grade, and Field Slope on Nutrient Losses in Runoff in Contour Ridge Systems under Seepage with Rainfall Condition

**DOI:** 10.3390/ijerph18042022

**Published:** 2021-02-19

**Authors:** Juan An, Jibiao Geng, Huiling Yang, Hongli Song, Bin Wang

**Affiliations:** 1Shandong Provincial Key Laboratory of Water and Soil Conservation and Environmental Protection, College of Resources and Environment, Linyi University, Linyi 276005, China; gengjibiao@126.com (J.G.); shl052610099@126.com (H.S.); 2Linyi Hydrology Bureau, Linyi 276005, China; carnationll@163.com; 3Jinyun Forest Ecosystem Research Station, School of Soil and Water Conservation, Beijing Forestry University, Beijing 100083, China

**Keywords:** seepage, ridge height, row grade, nutrient loss, rainfall simulation

## Abstract

Seepage plays a key role in nutrient loss and easily occurs in widely-used contour ridge systems due to the ponding process. However, the characteristics of nutrient loss and its influential factors under seepage with rainfall condition in contour ridge systems are still unclear. In this study, 23 seepage and rainfall simulation experiments are arranged in an orthogonal rotatable central composite design to investigate the role of ridge height, row grade, and field slope on Nitrate (NO_3_^−^–N) and Orthophosphate (PO_4_^+3^–P) losses resulting from seepage in contour ridge systems. In total, three types of NO_3_^−^–N and PO_4_^+3^–P loss were observed according to erosion processes of inter-rill–headward, inter-rill–headward–contour failure, and inter-rill–headward–contour failure–rill. Our results demonstrated that second-order polynomial regression models were obtained to predict NO_3_^−^–N and PO_4_^+3^–P loss with the independent variables of ridge height, row grade, and field slope. Ridge height was the most important factor for nutrient loss, with a significantly positive effect and the greatest contribution (52.35–53.47%). The secondary factor of row grade exerted a significant and negative effect, and was with a contribution of 19.86–24.11% to nutrient loss. The interaction between ridge height and row grade revealed a significantly negative effect on NO_3_^−^–N loss, whereas interactions among the three factors did not significantly affect PO_4_^+3^–P loss. Field slope only significantly affected NO_3_^−^–N loss. The optimal design of a contour ridge system to control nutrient loss was obtained at ridge height of 8 cm, row grade of 2°, and field slope of 6.5°. This study provides a method to assess and model nutrient loss, and improves guidance to implement contour ridge systems in terms of nutrient loss control.

## 1. Introduction

Transport of dissolved nutrients from agricultural fields is a significant cause of land degradation and aquatic environmental deterioration around the world, and results in a threat to the capacity of land to provide ecosystem services which are necessary requirement to achieve goals of the sustainable development goals (SDGs) [1,2]. Nutrient loss in runoff has increasingly become a threat to food security and water quality [3,4]. Nutrient losses of up to 81% for nitrogen and 93% for phosphorus have been discovered in runoff arising from agriculture-related activities in China [5]. Nitrate (NO_3_^−^–N) and orthophosphate (PO_4_^+3^–P) are regarded as the principal nutrients that cause the eutrophication of water bodies [6]. Seepage flow plays an important role as conveyers of nitrogen and phosphorus from fields to watercourses. Field and laboratory observations have revealed that NO_3_^−^–N and PO_4_^+3^–P concentrations and losses in seepage flow were significantly higher than those in surface run-off [7,8,9,10,11]. Therefore, it is urgent to study the process of nutrient loss under seepage condition to avoid further land degradation, which can apply the guidance for SDGs in the perspective of soil-water system solutions relate to water management.

Nutrient loss related to seepage flow is a dynamic and complex process. Recently, studies have primarily focused on nutrient loss under the two models of seepage occurrence. One model of soil banks is generated from pipe flows when a perched water table is above the restrictive layer [12,13]. In this model, seepage flow that occurs on hillslope is generally modeled in laboratory by supplying water from the upper side of experimental plot under a certain water head, and has been shown to trigger a significant portion of nutrient loss by decreasing soil strength [14,15]. The other model that applies to the downslope of slope land, seepage flow owns a positive hydraulic gradient that occurs under an exfiltration condition [16]. The exfiltration gradient that is against the gravitational flow has been found to reduce the effective stress between soil particles and increase soil erodibility [7,17,18]. To study nutrient loss at the downslope of slope land, seepage has normally created by supplying water from the bottom of experimental plots at higher water level relative to soil surface until seepage flowed out at steady state [19,20,21,22,23]. However, nutrient loss in another seepage generation models within contour ridge systems has been overlooked.

Contour ridge systems are used worldwide as an agronomic practice that improves grain yield and controls soil erosion [24,25,26,27,28,29]. In northern China, contour ridge system is the most effective tillage practices for sloped lands [30] (Figure 1a,b). Due to the irregular microtopographic relief, the ridges are much too hard to precisely follow the contours on slopes [31], and induces the formation of furrow depressions. During rainfall, rainwater from side slopes accumulates in these depressions (Figure 1c), namely the ponding process, which causes a low connectivity between inter-rows and favors the flow of water and eroded soil along the inter-rows [32]. Ponding results in rainwater to further infiltrate in the furrows and induces a difference in the water potential gradient between the furrow areas and ridges. When soil is saturated, the potential gradient leads seepage flowing from the side slopes of ridges, namely soil saturation pattern become the key driver for water connectivity [33] and then improves seepage occurrence. Additionally, eroded soil depositing in furrow depressions following ponding water (Figure 1d) decreases relative ridge height and then enhances water connectivity, and thus improves seepage occurrence. Thereby, seepage is a common phenomenon within contour ridge systems. Importantly, seepage flow has been found to be statistically higher than that of flat tillage and longitudinal ridge systems [34]. Therefore, nutrient loss under seepage condition should be of great concern within contour ridge systems.

Contour ridge systems consist of ridges, adjacent furrows, and implemented field topography, which induces soil surface roughness that plays a key role for runoff generation [35]. Therefore, the microtopography and ridge geometry are two of the primary factors that affect seepage and corresponding nutrient loss. Ridge height that is regarded as the most representative indices of the microtopography controls the length of row side-slope, and determines the amount of ponding water. Row grade is defined as the deviation angle of the contour ridge from the contour line, and is the most essential cause for seepage occurrence [16]. Ridge height and row grade are considered as sub-factors to the support practice factor (P) within contour ridge system in the Revised Universal Soil Loss Equation, Version 2 (RUSLE2) [36]. Previous studies have primarily focused on the effects of ridge height and row grade on the yielding process of surface runoff [27,37,38]. It was acknowledged that ridge height positively affected rainwater infiltration and then induced less runoff [36], and row grade exhibited significant influence on surface runoff [27,39]. In the RUSLE2 and the Water Erosion Prediction Project (WEPP) models, ridge height and row grade were used to calculate infiltration process and the amount of accumulated water in furrows [36,40]. Field slope that greatly reflects ridge geometry exhibits a significant influence on runoff occurrence [19], and has a convex curve shape for soil erosion in the RUSLE2 model [36]. In recent, a little information has been obtained on the effects of ridge height, row grade, and field slope on seepage generation [16,41]. However, few studies have linked nutrient loss to the roles and interactions between ridge height, row grade, and field slope, particularly under seepage with rainfall condition.

A more thorough understanding of the process of nutrient loss and its influential factors under seepage with rainfall condition in contour ridge systems will improve our knowledge of nutrient transport mechanism. Therefore, this study was to examine nutrient loss induced by seepage in contour ridge systems. The specific objectives were to: (i) analyze the characteristics of NO_3_–N and PO_4_^+3^–P losses in the combination of seepage and surface runoff; (ii) quantify the significant effects and contribution of ridge height, row grade, field slope on NO_3_^−^–N and PO_4_^+3^–P losses, and then identify the main factors; and (iii) determine the optimal design of factors for nutrient loss control.

## 2. Materials and Methods

### 2.1. Experimental Design

The orthogonal rotatable central composite design was employed to determine the effects of three influential factors on NO_3_^−^–N and PO_4_^+3^–P losses, namely, ridge height (RH), row grade (RG), and field slope (FS). The second-order response model for the design is expressed as: $Y=\beta_{0}+\sum_{i-1}^{k}\beta_{i}X_{i}+\sum_{i-1}^{k}\beta_{ii}X_{i}{}^{2}+\sum_{i-1}^{k-1}\sum_{j=i+1}^{k}\beta_{ij}X_{i}X_{j}$ (*Y* is the response variable; β is the coefficients of variables; *X* is the code values of independent variables). This model provides best precision evaluation of the effects and interactions of influential factors according to these uncorrelated estimates of the regression coefficients [42,43]. The “+” and “−” symbols in the model identify the effect being positive or negative, respectively [44]. Importantly, this design requires a smaller number of experiments compared with the full factorial design and orthogonal array, in which the test number of design points (*N*) included 2*^k^* factorial points, 2*k* axial points, and *C*_0_ enter points (where *k* is the number of influential factors). In our study, *C*_0_ value was settled at 9 to estimate experimental error and condition uniformity.

The minimum and maximum values for ridge height, row grade, and field slope at the corresponding code gradients of −1.682 and1.682 ($\sqrt[{4}]{2^{k}}$) were determined according to the results of in situ investigation and previous studies. Then, the values at the other code gradients (i.e., 1, 0, and −1) could be determined using Equation (1) (Table 1). A total of 23 treatments were obtained with different combinations of code gradients (Table 2). Among the treatments, treatments 1–14 were replicated two times. Thus, a total of 37 runs were randomly performed in this study.
(1)$$\left\{ \begin{array}{l} X_{j}=\frac{Z_{j}-Z_{0j}}{\Delta_{j}}\\ \Delta_{j}=\frac{Z_{2j}-Z_{1j}}{2\gamma }\\ \gamma =\sqrt[{4}]{2^{k}} \end{array}\right.$$
where *X_j_* is the coded value of influential factors; *Z_j_*, *Z*_0*j*_, and ∆*j* are the values of influential factors, the values of influential factors at the zero code gradient, and the step change, respectively; *Z*_2*j*_ and *Z*_1*j*_ are the maximum and minimum value of factors; *γ* is the asterisk arm; *k* is the number of factors.

### 2.2. Experimental Setup

#### 2.2.1. Rainfall Simulator System

Simulated rainfall experiments were performed using a trough rainfall simulator with veejet-80100 nozzles for the spraying system [45], which was available in the Shandong Provincial Key Laboratory of Soil Conservation and Environmental Protection, Linyi City, PR China. The simulator had a homogeneity coefficient of >0.89, and could be set to rainfall intensity ranging from 10 to 200 mm h^−1^ by adjusting the spray nozzle size and water pressure. The fall height of simulated rainfall system could be adjusted to the maximum height of 16 m above the ground, which matched the requirement for most raindrops to reach the terminal velocity. The simulated raindrop diameter ranged from 0.5 to 3.0 mm (raindrops size of 0.2–3.8 mm in the field), and more than 80% of raindrop diameters were smaller than 1.0 mm, which was similar to that in the field (more than half of raindrops being <1.2 mm). Prior to rainfall experiments, calibration of rainfall intensity spatial distribution was conducted to ensure that experimental rainfall intensity reached the designed rainfall intensity and evenly distributed over experimental plot.

#### 2.2.2. Experimental Soil

A sandy brown soil (USDA Taxonomy) that generated from granite was used in this study. The brown soil is widely distributed in rocky mountain areas of northern China, and seepage readily occurs in this soil on the sloping land within contour ridge systems. The soil was collected at a sampling depth of 0–20 cm from maize (*Zea mays* L) field near Fei County (118°17′59″ E, 35°19′27″ N), Linyi City, Shandong Province. The used soil contained 71.2% sand (2–0.05 mm), 28.1% silt (0.05–0.002 mm), and 0.7% clay (<0.002 mm). The soil contained 13.3 g kg^−1^ of soil organic matter, 1.2 g kg^−1^ total N, 15.0 mg kg^−1^ NO_3_^−^–N, and 1.0 g kg^−1^ total P. The soil pH in water was 5.0.

#### 2.2.3. Experimental Plots

A row slope and field slope-adjustable, 160-cm long, 160-cm wide, and 0.4-cm deep soil plot was used with holes (2 cm aperture) at the bottom (Figure 2) to simulate nutrient loss at hillslope scale. Row grade ranging from 0° to 15° was obtained by adjusting a screw (a), and field slope could be created from 0° to 20° by rotating the screw (b). The outlet (g) fixed in the middle of the downside of experimental plot to collect seepage and surface runoff samples.

The soil bed was packed by vertical layering with different bulk densities that simulated field observations of typical farmlands in the rocky mountain area of northern China. Prior to packing, the soil was passed through a 10.0 mm sieve after being air dried to avoid the influences of large gravel and plant materials on nutrient loss. The bottom layer of experimental plot was packed in four 5-cm layers with a soil bulk density of 1.6 g cm^−3^. Then, 192.18 g of fertilizer (N:P:K = 18:18:18) was introduced at the surface of the final soil layer. Finally, two ridges were built at a bulk density of 1.2 g cm^−3^ in accordance with the ridge geometry that was drawn on the plot walls above the bottom line of furrow. The ridge width was 80 cm, which is widely used for peanuts (*Arachis hypogaea*) and maize cultivation in the northern China. To maintain the packed soil being in the same weight for different ridge heights, the level of furrow bottoms was adjusted. For example, when the ridge volume was diminished for the case of decreased ridge height, the level of furrow bottom was increased to pack more soil. During packing process, when soil layer was compacted to the defined boundary level, the soil surface was lightly scratched with a wooden board to keep the uniformity and continuity of soil structure. After building contoured ridge systems, the soil beds were left untreated for 24 h.

### 2.3. Experimental Procedure

#### 2.3.1. Experimental Seepage Process

Seepage experiment was conducted to create a stable seepage condition prior to the simulated rainfall experiment. Prior to seepage experiment, a 60-min pre-rain with a rainfall intensity of 20 mm h^−1^ was implemented to the soil surface in order to consolidate the loose aggregates and keep consistent soil moisture. During this process, a nylon net with 1 mm aperture was placed 10 cm over the experimental plot to weaken raindrop impacts on infiltration and aggregate breakdown. One day after this pre-rain phase, clean tap water was pumped into the furrow at a discharge rate of 3 L min^−1^ through two pipes (c) with 2-cm diameters (Figure 2). Then, eight holes (each with a diameter of 1 mm) were drilled on the wall of each pipe. To supply gentle discharge, these holes and the terminal outlets of pipes were wrapped using gauze. During the water supply phase, the water level in the furrow was maintained 1 cm lower than the concave spot at the downslope ridge (h). While, the sinking of ridge would lead to the decrease in ridge height. To avoid overflow and achieve seepage requirement, the two pipes (d) that were fixed in the bottom-center of plot was manually adjusted to allow the excess water to drain through the tubes (f). When continuous seepage flowed out from the outlet (g), seepage flow was manually collected at 2-min intervals. Once seepage discharge reached a stable level, the water supply was stopped through closing the two pipes (d).

#### 2.3.2. Simulated Rainfall Process

Rainfall simulations at a rainfall intensity of 39 ± 0.4 mm h^−1^ were conducted for 30 min. For each rainfall event, once runoff flow occurred that included seepage and surface runoff, runoff samples were collected with 5-L buckets at one-minute intervals. The time at which a rill or contour failure occurred was noted. After rainfall, the collected runoff samples were immediately weighed, and then settled for approximately six hours to precipitate suspended sediments. A 100-mL sub-sample was collected from each bucket, and two samples for tap water were collected during each rainfall event. These samples were filtered through 0.45-um filter paper. NO_3_-N and PO_4_^+3^–P concentration in filtered runoff and tap water samples were determined within 24 h by the double wavelengths colorimetric method and the molybdenum blue spectrophotometry method using the continuous flow analyzer, respectively. Each rainfall event was conducted as an independent test, in which the bottom layer and ridge systems of experimental plots were reconstructed with the pre-treated soil prior to the next rainfall.

### 2.4. Data Analysis

The second-order polynomial regression models for NO_3_^−^–N and PO_4_^+3^–P losses were performed using the data processing system (DPS 7.05) [46]. Meanwhile, analysis of variance (ANOVA) in the DPS 7.05 software (Manufacturer, City, State abbreviation Zhejiang university, Hangzhou, China) was conducted to assess the significance of regression coefficients and the quality of model fit at the significant level of *p* < 0.05, and to determine the contribution level of factors.

## 3. Results

### 3.1. The Loss Characteristic of NO_3_^−^–N and PO_4_^+3^–P during Rainfall

The losses of NO_3_^−^–N and PO_4_^+3^–P occurred under various seepage discharges ranging from 0.25 to 1.27 L min^−1^ (Table 2). The loss of NO_3_^−^–N varied more considerably relative to PO_4_^+3^–P, with the maximum value of 6.53 g for No.5 and the minimum value of 0.91 g for No.13. A total of three types of loss characteristics for NO_3_–N and PO_4_^+3^–P were found in the 23 treatments based on erosion process. For the first type of treatments that experienced inter-rill (I) and headward erosion process (H) (Figure 3b), although concentration of NO_3_^−^–N and PO_4_^+3^–P (treatment No.13) displayed a gradual decreasing trend during inter-rill erosion process, their loss sharply increased to the maximum before continually decreasing (Figure 4a). This increase in nutrient loss is possibly induced by the increase in rainfall-generated runoff from the ridge side-slope at which the soil surface was saturated. The decrease was more likely caused by the decrease in seepage discharge after water supply being closed.

The second type of treatments after experiencing inter-rill and headward erosion, the ponding water in the furrow rushed down and then resulted in ridge collapse, which finally induced the formation of contour failure (Figure 3c). With the occurrence of contour failure that enhanced runoff connectivity between furrows and ridge slope, NO_3_^−^–N and PO_4_^+3^–P concentration reached a minimum (treatment No.16), but NO_3_^−^–N and PO_4_^+3^–P losses were, respectively, more than 130 mg min^−1^ and 9 mg min^−1^, which were 1.01–1.21 times higher than that during headward erosion (Figure 4b). Therefore, contour failure should be given greater attention. For the third type of treatments, a rill formed at the flat bottom of the side-slope with rill bank failure after contour failure (Figure 3d). During this phrase, due to the limited runoff, the losses of NO_3_–N and PO_4_^+3^–P (treatment No.21) reached a minimum value during the whole erosion processes (Figure 4c).

During inter-rill process, hug difference in NO_3_^−^–N and PO_4_^+3^–P losses existed among these three types of treatments. The second type of treatment No.16 with a 11.65-min duration resulted in the greatest nutrient loss, followed by the third type of treatment No.21 with a 7.25-min duration, and the first type of treatment No.13 with a four-minute duration (Figure 4). These results indicated that nutrient loss increased with an increase in the duration of inter-rill erosion, which was similar to that during headward erosion process. Namely, the duration of inter-rill process affected nutrient loss prior to the occurrence of contour failure, which could be attributed to the temporal change in seepage rate. With an increase in duration, the water level increased continually before overflow occurred and thus caused a higher gradient between furrows and ridges, which could enhance seepage discharge and correspondingly increase nutrient loss. However, the effect of duration became negligible after contour failure as evidenced by that no pronounced difference was observed between the second and third types during contour failure process (Figure 4b,c). This could be explained by the fact that a large amount of ponding water rushed down and even disappeared from the furrows, along with the occurrence of contour failure, which sharply reduced seepage rate. Therefore, focusing on seepage rate during erosion process can improve the prediction of nutrient loss.

### 3.2. Effects and Interactions of the Influencing Factors on NO_3_^−^–N and PO_4_^+3^–PLosses

#### 3.2.1. The Second-Order Polynomial Regression Models for NO_3_^−^–N and PO_4_^+3^–P Losses

Using DPS 7.05 software with the orthogonal rotatable central composite design of statistical method, the second-order polynomial regression model (2) for NO_3_^−^–N loss and model (3) for PO_4_^+3^–P loss were constructed with the code values of ridge height (X_1_), row grade (X_2_), and field slope (X_3_) as the independent variables. ANOVA analysis showed the *p*-value for the lack of fit was larger than 0.05 for the two models (Table 3), indicating other unknown factors had slightly influence on the accuracy of experimental results. In addition, ANOVA analysis indicated that models (2) and (3) reached an extreme significance level according to the very small *p*-values of 0.00. Therefore, it can be concluded that models (2) and (3), respectively, had a good fit for NO_3_^−^–N and PO_4_^+3^–P loss. Importantly, the predicted value for NO_3_^−^–N loss and PO_4_^+3^–P loss, respectively, matched better with the observed value (Figure 5a1,b1). Moreover, the relationship between predicted and observed NO_3_^−^–N and PO_4_^+3^–P loss, respectively, followed the 1:1 line, as shown in Figure 5a2,b2. Therefore, models (2) and (3) adequately reflected the actual circumstance of NO_3_^−^–N and PO_4_^+3^–P loss based on the values of ridge height, row grade, and field slope, respectively.
(2)$$Y_{N}=2.76+1.01X_{1}-0.62X_{2}+0.25X_{3}+0.05X_{1}{}^{2}-0.03X_{2}{}^{2}+0.25X_{3}{}^{2}-0.65X_{1}X_{2}+0.24X_{1}X_{3}-0.4X_{2}X_{3}$$
(3)$$Y_{P}=418.58+137.22X_{1}-97.15X_{2}-25.47X_{3}+53X_{1}{}^{2}-21.13X_{2}{}^{2}-33.04X_{3}{}^{2}-53.3X_{1}X_{2}-24.09X_{1}X_{3}+55.15X_{2}X_{3}$$

#### 3.2.2. The Effects of Influential Factors on NO_3_^−^–N Loss

##### The Significant Effects and Contribution of Factors on NO_3_^−^–N Loss

The influence magnitudes of ridge height, row grade, and field slope on NO_3_^−^–N loss can be sufficiently explained by the regression coefficients in the fitting models (2). As for the factor effect, the regression coefficient of ridge height was greater than that of row grade and field slope, meaning that ridge height exerted the greatest influence on NO_3_^−^–N loss. Ridge height exhibited a significant and positive affect, and row grade showed a significantly negative effect (*p* < 0.05) according to the corresponding *p*-values and “+” or “−” symbols in the model (Table 3). Field slope exhibited no significant effects, but its quadratic term showed a significant and positive effect. In aspect of the interaction effects, ridge height_*_row grade had a greater influence according to its larger regression coefficients, followed by row grade_*_field slope and ridge height_*_field slope. Row grade and its interaction with ridge height and field slope significantly and negatively affected NO_3_^−^–N loss, indicating that NO_3_^−^–N loss exhibited smaller increase rate under large row grade on higher ridges and steeper slopes. Whereas, the interaction of ridge height_*_field slope exerted a positive effect, albeit not significantly. Importantly, the regression coefficient of ridge height was larger than that of ridge height_*_row grade.

To further determine the influence degree of factors on NO_3_^−^–N loss, the level of contributions of factors was analyzed. Among these, ridge height contributed most to NO_3_^−^–N loss, with a contribution of 52.35% (Figure 6a). Then, row grade contributed 19.86% to NO_3_^−^–N loss, which was trailed the interaction of ridge height_*_row grade with a contribution of 11.63%. While, the other factors contributed less than 6% to NO_3_^−^–N loss. Therefore, according to the significant effects and contribution, ridge height was the most striking factor affecting NO_3_^−^–N loss, followed by row grade, and their interaction.

##### The Main Factor for NO_3_^−^–N Loss Response

To determine how the main factor of ridge height and row grade affect NO_3_^−^–N loss, their monofactor effect were discussed in Figure 7 by setting the other two factors at zero level from model (2). Although the monofactor effects of ridge height and row grade were quadratic equations, NO_3_^−^–N loss gradually increased with the increase in ridge height, whereas row grade showed inverse effects, NO_3_^−^–N loss continually decreasing. Namely, the effects of ridge height and row grade became a linear function because the quadratic terms in model (2) were insignificant. This indicated that ridge height and row grade exhibited one-way effect on NO_3_^−^–N loss.

The significant and main interactions effects of ridge height_*_row grade on NO_3_^−^–N loss were presented in Figure 8 by setting field slope to zero from model (2). At a certain code value of ridge height, NO_3_^−^–N loss gradually increased with the increase in row grade when its code value was smaller than −0.51, and then exhibited a decreasing trend. This means that using fixed-ridge height plots to discuss the effect of row grade on NO_3_^−^–N loss would give different results. For a given row grade, NO_3_^−^–N loss continually increased as the increase in ridge height, and exhibited a greater increased gradient when the code value of ridge height was larger than zero. This indicated that NO_3_^−^–N control could reach the optimal condition under a certain combination of ridge height and row grade.

#### 3.2.3. The Effects of Various Factors on PO_4_^+3^–P Loss

##### The Significant Effects and Contribution of Various Factors on PO_4_^+3^–P Loss

Based on the regression coefficients in model (3) for PO_4_^+3^–P loss, the single factor effect decreased in the order of ridge height, row grade, and field slope. The low *p* value (0.00) indicated that ridge height and row grade exhibited a significant influence (Table 3). Ridge height positively affected PO_4_^+3^–P loss, but row grade had a negative effect. In addition, the interactions among these three factors were not significant, even at *p* < 0.05. This indicated that ridge height and row grade dominated PO_4_^+3^–P loss, and the interactions could be ignored. This can be further proved by the contribution of factors. Ridge height contributed the greatest to PO_4_^+3^–P loss, with a contribution of 53.47%, which was followed by row grade that contributed 24.11% to PO_4_^+3^–P loss (Figure 6b). While, the other factors made contributions of <3.5% to PO_4_^+3^–P loss. Therefore, control measures for PO_4_^+3^–P loss related to key factors of ridge height and row grade could be individually adopted in contour ridge system.

##### The PO_4_^+3^–P Loss Response to the Key Factors

The response of PO_4_^+3^–P loss to ridge height and row grade was depicted in Figure 9 by the other two factors at zero level from model (3). Figure 9 shows that, with increasing ridge height, PO_4_^+3^–P loss initially decreased and then gradually increased. This indicated that ridge height exhibited a two-way effect on PO_4_^+3^–P loss, namely a one-way negative effect at code values lower than 0 and then a one-way positive effect. In addition, PO_4_^+3^–P loss gradually declined with the code value of row grade, indicating that row grade only exerted a one-way negative effect on PO_4_^+3^–P loss. Therefore, PO_4_^+3^–P loss can be effectively controlled at a certain ridge height and row grade.

### 3.3. The Optimal Design of Factors for the Control of NO_3_^−^–N and PO_4_^+3^–P Losses

#### 3.3.1. The Optimal Design for NO_3_^−^–N Loss Control

Based on the statistics frequency analysis method, when the design combinations of ridge height, row grade, and field slope at five levels (125 treatments) were taken in model (2), 57 treatments of NO_3_^−^–N loss were found to have a greater than the average value of 2.89 g. When the code value of ridge height, row grade, and field slope were, respectively, in the range of 0.592–1.032 (natural value of 13.42–14.48 cm), −0.768–0.197 (natural value of 4.16–5.53°), and −0.484–0.213 (natural value of 8.55–10.64°), 95% of NO_3_^−^–N loss was greater than the mean within these 57 treatments (Table 4). Simultaneously, the loss of NO_3_^−^–N reached the maximum at ridge height of 16 cm, row grade of 2°, and field slope of 15°. Therefore, NO_3_^−^–N loss can be rather effectively controlled at ridge height of 8.0–13.3 cm and 14.5–15.9 cm, row grade of 2.1–4.1° and 5.6–10.0°, and field slope of 5.0–8.4° and 10.7–14.9° in contour ridge systems. By combining the suitable ridge condition and the prediction model (2), model (4) was established. Based on model (4), the minimum NO_3_^−^–N loss (0.12 g) was obtained at ridge height of 8 cm, row grade of 2.1°, and field slope of 6.5° using the revised simplex acceleration method, which was decreased by 95.85% related to the average of the 23 treatments.
(4)$$\left\{ \begin{array}{l} M{\mathrm{inY}}_{N}=2.76+1.01X_{1}-0.62X_{2}+0.25X_{3}+0.05X_{1}{}^{2}-0.03X_{2}{}^{2}+0.25X_{3}{}^{2}-0.65X_{1}X_{2}+0.24X_{1}X_{3}-0.40X_{2}X_{3}\\ -1.682<X_{1}<0.592\;\;1.032<X_{1}<1.682\\ -1.682<X_{2}<-0.768\;\;0.197<X_{1}<1.682\\ -1.682<X_{3}<-0.484\;\;0.213<X_{1}<1\\ Y_{N}>0 \end{array}\right.$$

#### 3.3.2. The Optimal Design for PO_4_^+3^–P Loss Control

When the combinations of ridge height, row grade, and field slope at the five levels (125 treatments) were substituted into model (3), 75 treatments had PO_4_^+3^–P losses greater than the mean (417.40 mg). Among these treatments, 95% of PO_4_^+3^–P loss occurred at the code value of ridge height of 0.412–0.893 (natural value of 12.99–15.14 cm), row grade of −0.698–0.179 (natural value of 4.32–6.43°), and field slope of −0.382–0.167 (natural value of 8.85–10.50°) (Table 4). In addition, the maximum of PO_4_^+3^–P loss occurred at ridge height of 16 cm, row grade of 2°, and field slope of 5.3°. Therefore, it can be concluded that PO_4_^+3^–P loss could be comparatively controlled at ridge height of 8.0–12.9 cm and 15.2–15.9 cm, row grade of 2.1–4.2° and 6.5–10.0°, field slope of 5.4–8.7° and 10.6–15°. Importantly, the revised simplex acceleration method showed that the minimum PO_4_^+3^–P loss (66.37 mg) occurred at ridge height of 8 cm, row grade of 2°, and field slope of 15° under the constraint condition in model (5). Under this optimal design condition, PO_4_^+3^–P loss was reduced by 84.10% as compared to the mean of the 23 treatments.
(5)$$\left\{ \begin{array}{l} M{\mathrm{inY}}_{P}=418.58+137.22X_{1}-97.15X_{2}-25.47X_{3}+53X_{1}{}^{2}-21.13X_{2}{}^{2}-33.04X_{3}{}^{2}-53.3X_{1}X_{2}-24.09X_{1}X_{3}+55.15X_{2}X_{3}\\ -1.682<X_{1}<0.412\;\;0.893<X_{1}<1.682\\ -1.682<X_{2}<-0.698\;\;0.179<X_{1}<1.682\\ -1.682<X_{3}<-0.382\;\;0.167<X_{1}<1.682\\ Y_{P}>0 \end{array}\right.$$

## 4. Discussion

### 4.1. The Effects of the Primary Factors on NO_3_^−^–N and PO_4_^+3^–P Losses

In ridge tillage practice, it is difficult to maintain uniform sizes of ridge geometry (ridge height) and contour microtopography (row grade and field slope) when contour ridges are constructed by the tractor-driven small machines or manually with simple tools. Various contour ridge geometries and microtopographies affecting soil surface roughness, which is prone to alter soil surface hydrology condition, and eventually shows influence on nutrient runoff [47], have been found to supply suitable growth conditions for different crops [48]. Therefore, the factors of ridge height, row grade, and field slope are likely to affect soil erosion [36,41], and may show various influence degrees for nutrient losses [11]. However, until recently, little information has been available regarding the effects on nutrient losses induced by ridge height, row grade, and field slope. Importantly, seepage more easily occurs in contour ridge systems [34], which was the very important transport way for nutrient loss [9,49]. In addition, seepage exhibited a larger effect on the flow-hydraulic through increasing flow velocity and shear stress, which could enhance nutrient loss [50]. Thus, it is urgent to determine the primary factors to effectively control nutrient losses in contour ridge systems experiencing seepage conditions.

In this study, the influence of design factors on NO_3_^−^–N and PO_4_^+3^–P losses decreased in the sequence of ridge height, row grade, and field slope based on the regression coefficients in models (2 and 3) and contribution levels. Ridge height, row grade, and their interaction exhibited a significant effect on nutrient loss (Table 3), while field slope exerted no significant effects. Under seepage without rainfall condition, An et al. [11] reported that ridge height significantly controlled NO_3_^−^–N and PO_4_^+3^–P loss, and row grade only significantly affected PO_4_^+3^–P loss. They also noted that the interaction between ridge height and row grade had insignificant influence on nutrient loss. In addition, Liu et al. [51] reported that ridge height and row grade could be used to estimate the seepage discharge. Under free drainage condition, Liu et al. [27] noted that ridge height significantly affected runoff relative to row grade and field slope, meaning that nutrient loss was primarily affected by ridge height. Therefore, it can be concluded that ridge height exhibited the greatest influence on nutrient loss both under seepage and free drainage conditions, followed by row grade. In tillage practice, adjusting ridge height and row grade can be more effective to control nutrient loss in runoff in contour ridge systems.

It was well acknowledged that an increase in ridge height results in a greater amount of water storage in furrows [36] and a higher water gradient, thus inducing a rapid increase in seepage discharge. In addition, a higher ridge height causes an increase in the row-side slope length and steepness [27]. Thus, nutrient loss dramatically increased as the increase in ridge height (Figure 6 and Figure 8). Namely, ridge height had a positive linear effect, which was inconsistent with the results of An et al. [11] under seepage without rainfall condition. They found that nutrient loss was improved continuously at a non-linear increasing rate with the increase in ridge height. For row grade, a slight row grade can result in serious ephemeral gully erosion [36], and contouring rapidly loses the effectiveness with an increase in row grade. A greater row grade that can decrease soil water capacity allows more water to flow from soil [40]. So, the increase in row grade can rapidly enhance nutrient losses. Under free drainage condition, Liu et al. [27] observed that row grade positively affected runoff during inter-rill and rill erosion process through simulated rainfall experiments, which was similar to that occurred during the phrase of contour failure [52]. While, Liu et al. [53] noted that row grade showed no obvious effect on runoff from concentrated flow in furrows of contour ridge systems. Unlike these previous studies, under free drainage conditions, row grade had one-way negatively effect on nutrient loss (Figure 8 and Figure 9) in this study, which also differed from the observations of An et al. [11] under seepage without rainfall condition. They reported that the effect of row grade was described as a concave curve, and row grade exerted a two-way effect. Therefore, the effects of ridge height and row grade on nutrient loss should be considered separately when the soil surface water regime is considered.

### 4.2. The Optimal Design of Contour Ridge System for Controlling Nutrient Loss

To obtain the optimal design of a contour ridge system under seepage with rainfall condition, this study firstly discussed the suitable ridging condition for the control of NO_3_^−^–N and PO_4_^+3^–P losses based on the statistics frequency analysis method. The suitable ridging condition occurred at ridge height of 8.0–12.9 cm and 15.2–15.9 cm, row grade of 2.1–4.1° and 6.5–10.0°, and field slope of 5.4–8.4° and 10.7–14.9° (Table 4). Under the basis of a suitable ridging condition, the minimum NO_3_^−^–N and PO_4_^+3^–P losses occurred at the same ridge height of 8 cm and row grade of 2° according to the revised simplex acceleration method, which were much smaller than those under seepage without rainfall and free drainage conditions. Under seepage without rainfall condition, Liu et al. [16] noted that the minimum seepage flow occurred at ridge height of 8.1 cm and row grade of 6.5°, while An et al. [11] reported that the optimized combination for controlling nutrient loss was at ridge height of 9.26 cm and row grade of 7.05°. Under free drainage condition, Liu et al. [27] noted that runoff at ridge height of 10 cm was significantly lower than that at ridge height of 15 cm. The discrepancy was possible because under seepage condition, nutrient loss was more sensitive to ridge height and row grade. However, the minimum NO_3_^−^–N and PO_4_^+3^–P loss occurred at quite different field slopes, field slope of 6.5° for NO_3_^−^–N loss and 15° for PO_4_^+3^–P loss. Whereas, field slope exhibited a positive effect on runoff when it shifted from 4.9° to 10.9° under seepage and free drainage conditions [27,36]. Additionally, seepage flow was lower at field slope of 5° compared with that at field slope of 10°, 15°, and 20° [54], and the soil and water conservation efficiency of contour ridge system was easily to be weakened under steep slopes [55]. Importantly, field slope exhibited no significant influence on PO_4_^+3^–P loss in this study. Therefore, 6.5° was reasonable for the optimized field slope.

Based on the above analysis, a combination of ridge height of 8 cm, row grade of 2°, and field slope of 6.5° was considered as the optimized design for a contour ridge system to control NO_3_^−^–N and PO_4_^+3^–P losses under seepage with rainfall condition. However, this optimal design requires further verification in field because nutrient loss was studied at a plot scale in our experiment. Importantly, to improve rainwater collection and soil conservation, modifications have been applied in contour ridge system, such as double furrows with raised beds [56], plastic film mulching [57], the use of catch crops [58], tied ridge [59]. Moreover, changes to the drainage layout have been considered to reduce seepage occurrence to further control nutrient loss, such as leading the water flow to converge in the form of a “V”, or to shunt in an inverted “V” [60]. Therefore, the combination the optimal design and modification or the use of drainage system can be more effective to control nutrient loss within contour ridge system under seepage condition. However, recently, the increasingly use of large tractor-driven machines, and the combination application of contour ridge system with other conservation measures, such as terrace and bench, resulted in varied microtopographies and ridge geometries. Thus, these stimulated factors (ridge height, row grade, and field slope) have not been completely reflected the real field circumstances. In this study, seepage experiments were performed under rainfall intensity of 39 mm h^−1^. If rainfall intensity becomes higher that shortens the duration of inter-rill erosion, seepage discharge may more sharply decrease to a lower rate, and then result in lower nutrient loss. Therefore, a wider scope of indices for the microtopographies, ridge geometries, and rainfall intensities could provide a basic guide for the improving use of contour ridge systems.

## 5. Conclusions

In total, 23 seepage and simulated rainfall experiments using the orthogonal rotatable central composite design were conducted to investigate NO_3_^−^–N and PO_4_^+3^–P loss process and its influential factors (i.e., ridge height, row grade, and field slope) under seepage with rainfall condition in contour ridge systems. Results showed that the process of nutrient losses could be divided into three types: I-H, I-H-C, and I-H-C-R (I, inter-rill; H, headward; C, contour failure; R, rill erosion stage), and inter-rill erosion periods produced the largest nutrient loss. The second-order polynomial regression models were obtained to predict nutrient losses. The dominated factor of ridge height exhibited a nearly linear significant and positive effect on nutrient loss, and was with a contribution of 52.35–53.47%. Row grade as the secondary factor revealed a one-way significantly negative effect and contributed 19.86–24.11% to nutrient loss. Field slope only exhibited a significant and concave curve effect on NO_3_^−^–N loss. Row grade had a significant and negative interaction with ridge height and field slope on NO_3_^−^–N loss, while interactions exerted insignificant effects on PO_4_^+3^–P loss. The optimal design for contour ridge system was determined at ridge height of 8 cm, row grade of 2°, and field slope of 6.5°.

## Figures and Tables

**Figure 1 ijerph-18-02022-f001:**
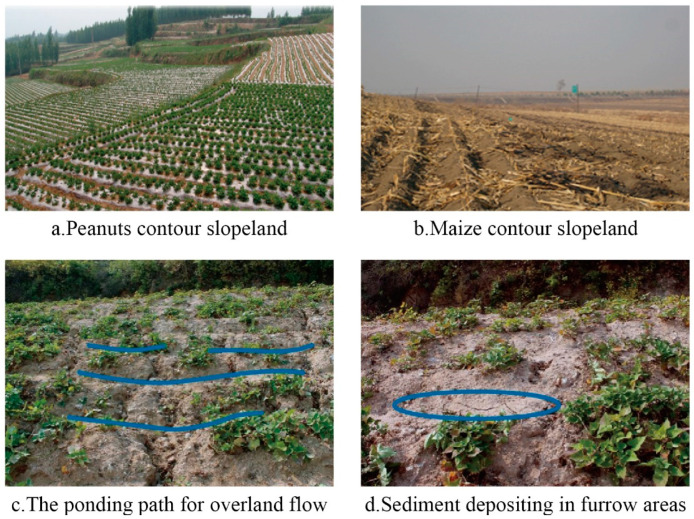
The application of contour ridge systems and erosion characteristics within contour ridge systems.

**Figure 2 ijerph-18-02022-f002:**
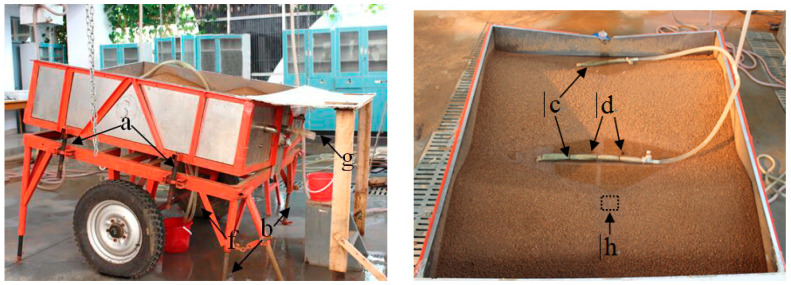
Experimental plot for adjusting row grade and field slope simultaneously by rotating a screw (a) and (b), and creating seepage condition by supplying water with a flow rate of 3 L min^−1^ to the furrows where the water level was controlled by pipes (d). a, screw for adjusting row grade; b, screw for adjusting field slope; c, pipes for supplying water; d, pipes for adjusting the water level 1 cm lower than the concave spot at the ridge (h); f, pipes for redundant water collection; g, outlet for collecting seepage and surface runoff.

**Figure 3 ijerph-18-02022-f003:**
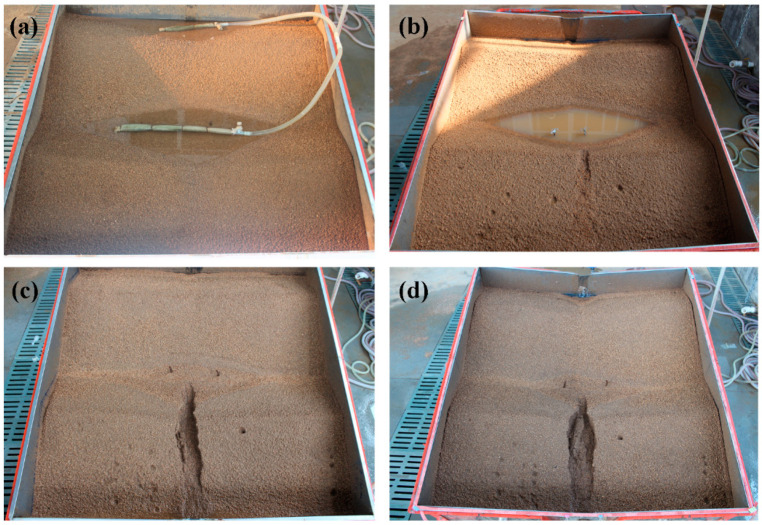
Soil surface morphology before and after rainfall experiments. (**a**) before rainfall; (**b**) inter-rill and headward erosion process; (**c**) contour failure; (**d**) rill erosion.

**Figure 4 ijerph-18-02022-f004:**
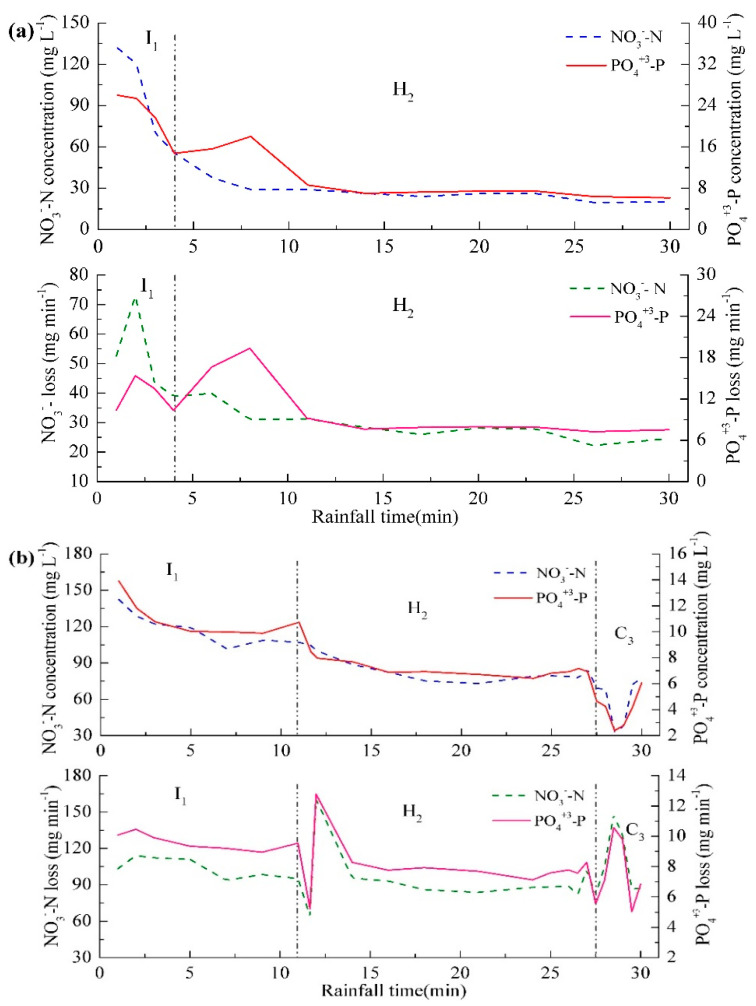
The temporal variation of NO_3_^−^–N and PO_4_^+3^–P concentration and loss during rainfall for (**a**) treatment No.13, (**b**) treatment No.16, and (**c**) treatment No.21. I_1_, inter-rill erosion; H_2_, headward erosion; C_3_, contour failure; R_4_, rill erosion.

**Figure 5 ijerph-18-02022-f005:**
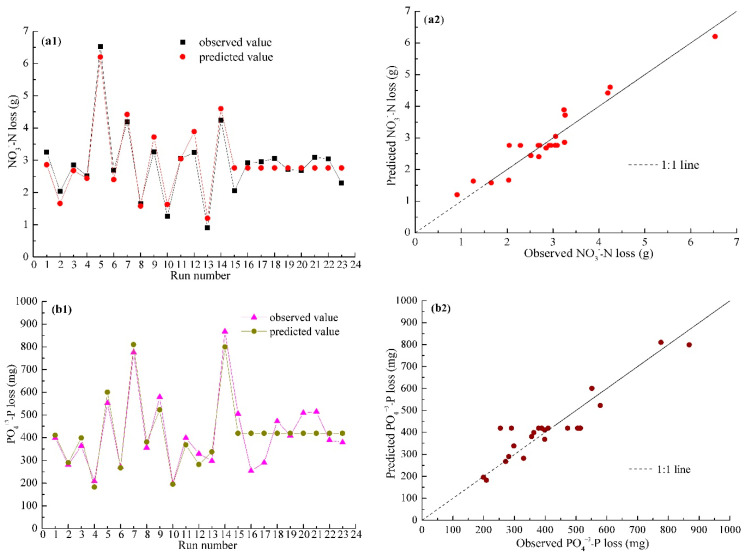
The curve and plot of the observed and predicted NO_3_^−^–N (**a1**,**a2**) and PO_4_^+3^–P loss (**b1**,**b2**).

**Figure 6 ijerph-18-02022-f006:**
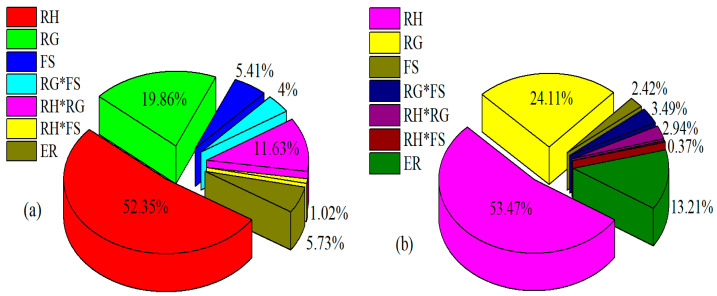
Contribution of influencing factors on NO_3_^−^–N loss (**a**) and PO_4_^+3^–P loss (**b**).

**Figure 7 ijerph-18-02022-f007:**
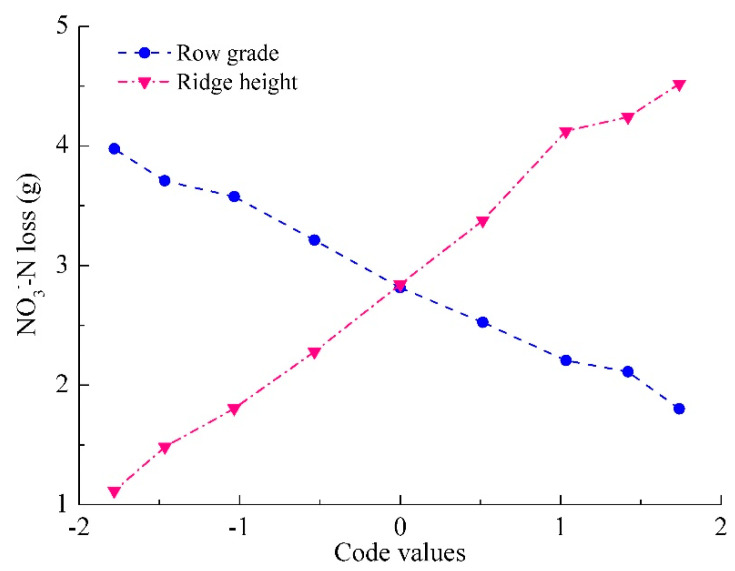
Mono-factor effect curves of ridge height and row grade for NO_3_^−^–N loss.

**Figure 8 ijerph-18-02022-f008:**
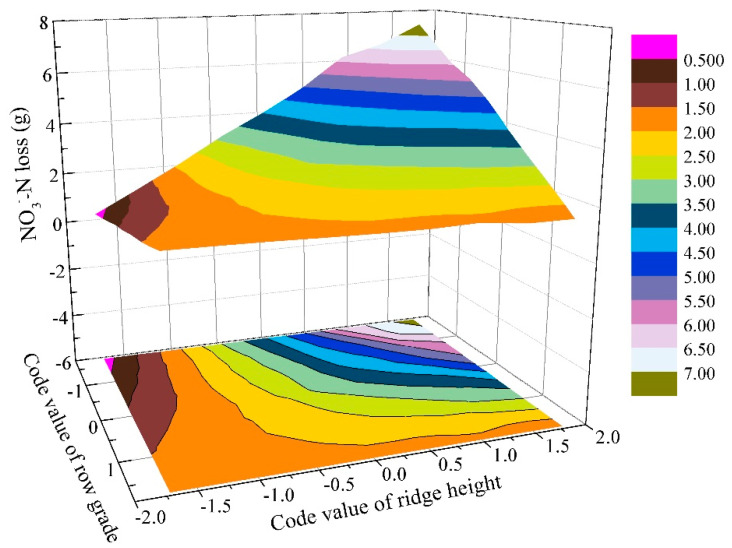
Interaction effects of ridge height_*_row grade on NO_3_^−^–N loss.

**Figure 9 ijerph-18-02022-f009:**
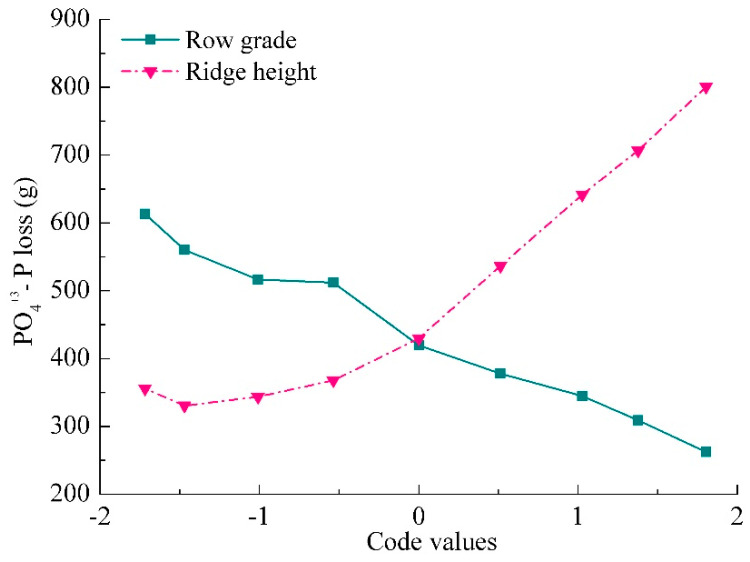
Mono-factor effect curves of ridge height and row grade for PO_4_^+3^–P loss.

**Table 1 ijerph-18-02022-t001:** Code gradients of ridge height, row grade, and field slope determined by an orthogonal rotatable central composite design and corresponding factors values.

Factors	Code Gradients				
	1.682	1	0	−1	−1.682
Ridge height (cm)	16	14.4	12	9.6	8
Row grade (°)	10	8.4	6	3.6	2
Field slope (°)	15	13	10	7	5

**Table 2 ijerph-18-02022-t002:** The orthogonal rotatable central composite design for ridge height, row grade, field slope, and measured values for seepage discharge and NO_3_^−^–N and PO_4_^+3^–P losses under various erosion types.

No	Code Gradients			Seepage Discharge * (L min^−1^)	Nutrient Loss (g)		Erosion Type ^#^
	Ridge Height (*X*_3_)	Row Grade (*X*_1_)	Field Slope (*X*_2_)		NO_3_^−^–N	PO_4_^+3^–P	
1	1	1	1	0.96	3.25	0.40	I-H-C-R
2	−1	1	1	0.78	2.04	0.28	I-H
3	1	1	−1	0.87	2.85	0.36	I-H
4	−1	1	−1	0.63	2.52	0.21	I-H
5	1	−1	1	1.04	6.53	0.55	I-H
6	−1	−1	1	0.69	2.69	0.27	I-H
7	1	−1	−1	0.80	4.19	0.78	I-H
8	−1	−1	−1	0.47	1.66	0.36	I-H
9	0	−1.682	0	0.93	3.26	0.58	I-H-C
10	0	1.682	0	0.90	1.26	0.20	I-H
11	0	0	−1.682	0.66	3.06	0.40	I-H
12	0	0	1.682	0.49	3.24	0.33	I-H
13	−1.682	0	0	0.25	0.91	0.30	I-H
14	1.682	0	0	1.27	4.24	0.87	I-H
15	0	0	0	0.52	2.05	0.51	I-H-C-R
16	0	0	0	0.62	2.92	0.25	I-H-C
17	0	0	0	0.35	2.95	0.29	I-H-C
18	0	0	0	0.82	3.06	0.47	I-H
19	0	0	0	0.59	2.71	0.41	I-H-C-R
20	0	0	0	0.62	2.68	0.51	I-H-C-R
21	0	0	0	0.79	3.09	0.51	I-H-C-R
22	0	0	0	0.61	3.05	0.39	I-H
23	0	0	0	0.42	2.29	0.38	I-H-C-R

* The average of seepage volume per min during the final 6 min of seepage experiment before rainfall. ^#^ I, inter-rill erosion; H, headward erosion; C, contour failure; R, rill erosion.

**Table 3 ijerph-18-02022-t003:** The regression coefficient significance test for the second-order polynomial. Regression models of NO_3_^−^–N and PO_4_^+3^–P losses.

Source *	NO_3_^−^–N Loss (g)			PO_4_^+3^–P Loss (mg)		
	Regression Coefficient	*F*-Value ^#^	*p*-Value	Regression Coefficient	*F*-Value	*p*-Value
*X* _1_	1.01	76.14	0.00	137.22	36.20	0.00
*X* _2_	−0.62	27.23	0.00	−97.15	19.05	0.00
*X* _3_	0.25	4.32	0.06	−25.474	1.25	0.29
*X* _1_ ^2^	0.05	0.19	0.73	53.00	6.60	0.03
*X* _2_ ^2^	−0.03	0.09	0.80	−21.13	1.03	0.35
*X* _3_ ^2^	0.25	5.09	0.04	−33.04	2.56	0.16
*X* _1*_ *X* _2_	−0.65	18.01	0.00	−53.30	3.36	0.11
*X* _1*_ *X* _3_	0.24	2.42	0.16	−24.09	0.70	0.47
*X* _2*_ *X* _3_	−0.40	6.61	0.03	55.15	3.68	0.10
Regression	-	14.81	0.00	-	7.91	0.00
Lack of Fit	-	1.89	0.21	-	0.45	0.82

* X_1_, the coded ridge height; X_2_, the coded row grade; X_3_, the coded field slope.^#^ F, the ratio of mean square to residual mean square.* denotes interaction.

**Table 4 ijerph-18-02022-t004:** The frequency distribution of ridge height, row grade, and field slope at 95% confidential interval for NO_3_^−^–N and PO_4_^+3^–P losses greater than the average.

Nutrient	Code Value			Natural Value		
	Ridge Height	Row Grade	Field Slope	Ridge Height (cm)	Row Grade (°)	Field Slope (°)
NO_3_^−^–N	0.592–1.032	−0.768–0.197	−0.484–0.213	13.42–14.48	4.16–5.53	8.55–10.64
PO_4_^+3^–P	0.412–0.893	−0.698–0.179	−0.382–0.167	12.99–15.14	4.32–6.43	8.85–10.50

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
