# Peer review of "Effect of Ridge Height, Row Grade, and Field Slope on Nutrient Losses in Runoff in Contour Ridge Systems under Seepage with Rainfall Condition"

_ijerph, 2021, doi:10.3390/ijerph18042022_

Round 1
Reviewer 1 Report
The prevention of biogenic losses from arable soils is a significant problem of modern agriculture around the world in terms of production and the environment.
The model studies on the selection of optimal contour ridge systems parameters presented in the reviewed article fit into this problem.
The authors exhaustively presented the method of conducting the research and the interpretation of the obtained results. It should be emphasized that advanced statistical methods were used for the analysis and interpretation of the obtained results. Therefore, I recommend that the article be accepted for printing without any corrections.
Author Response
Thanks for your suggestion.
Reviewer 2 Report
The introduction and discussion must be improved
Pay attention to the recent literature and improve the figures
see my comments attached

Author Response
1.The introduction and discussion must be improved. Pay attention to the recent literature and improve the figures.
Response: As suggested, information for the sustainable development goals, connectivity, roughness have been added in “introduction” section in the revised manuscript. In addition, recent information for soil surface roughness, the effect of seepage on flow hydraulic, the effect of row grade on seepage and runoff, the effect of field slope on soil loss, solutions to reduce seepage and soil loss in contour ridge system have been added in “discussion” section in the revised manuscript. These recently corresponding literatures have been referenced in the revised manuscript.
- your research is relevant to achieve the sustainable development goals of the United nations and the land degradation neutrality challenge. You should mention this to power your research and strength your rationale see here the research of professor Keesstra, S., Mol, G., de Leeuw, J., Okx, J., de Cleen, M., & Visser, S. (2018). Soil-related sustainable development goals: Four concepts to make land degradation neutrality and restoration work. Land, 7(4), 133.
Visser, S., Keesstra, S., Maas, G., & De Cleen, M. (2019). Soil as a Basis to Create Enabling Conditions for Transitions Towards Sustainable Land Management as a Key to Achieve the SDGs by 2030. Sustainability, 11(23), 6792.
Response: As suggested, we have carefully read the two references and have added the information of the land degradation neutrality in the revised manuscript (L36-38, L47-48).
- Line 76 You need to mention here the idea of connectivity as this is relevant to understand the process you describe see here a key paper. Keesstra, S., Nunes, J. P., Saco, P., Parsons, T., Poeppl, R., Masselink, R., & Cerdà, A. (2018). The way forward: can connectivity be useful to design better measuring and modelling schemes for water and sediment dynamics? Science of the Total Environment, 644, 1557-1572.
Response: As suggested, we checked these corresponding references for connectivity. Some information for connectivity has been added in the revised manuscript (L71-72, L75-76, L78).
- Line 141, Please mention the use of runoff simulators such as this paper shows. Which is very interesting for your research too.
Response: In our study, simulated rainfall experiments were conducted to analyze the effects of ridge height, row grade, and field slope on nutrient loss, not using flow experiments. Therefore, runoff simulator was not mentioned in our study. While, experimental plots which was used to simulate nutrient loss in seepage and surface flow (runoff) on hillslope was mentioned in the section “2.2.3. Experimental plots”.
- Line148, show the reader the rainfall simulators are of interest for different objectives in soil science. Zhao, L., Hou, R., Wu, F., & Keesstra, S. (2018). Effect of soil surface roughness on infiltration water, ponding and runoff on tilled soils under rainfall simulation experiments. Soil and Tillage Research, 179, 47-53.
Response: As suggested, we carefully read the reference (Zhao et.al, 2018), especially for the section “2.4.1. Rainfall simulator and procedure”. To more clearly describe the rainfall simulator, the corresponding information for the fall height of simulated rainfall system and the distribution of raindrops size have been added in the revised manuscript (L193-198).
- The graphs (Figure 4) a ok but they need to be aligned the x axis of the left are too many.
Response: In our study, a total of 23 treatments were conducted with different combinations of ridge height, row grade, and field slope. To comprehensively and clearly compare the difference between the observed value and predicted value for nutrient loss for all treatments, the run number of the total 23 treatments was kept in the x axis.
- I suggest a table instead of two graphs (Figure 5) Easy to read for the reader.
Response: Table may be also easy for readers to distinguish the difference in the contribution of influencing factors on nutrient loss. While, we believed that it is more clearly and directly to compare the contribution of influencing factors on nutrient loss in the form of figures.
- Line 369, space?
Response: The space between contents and Figure.8 has been added in the revised manuscript.
- Line 483, Use recent publications for the discussion... there are many state of the art that will help
Response: As suggested, recent references that were related to soil surface roughness, the effect of seepage on flow-hydraulic, the effect of row grade on seepage and runoff, the effect of field slope on soil loss, solutions to reduce seepage and soil loss in contour ridge system have been added in “discussion” section in the revised manuscript.
- in the last part of your discussion you must check the findings of other authors and also to show that there are solutions to reduce the nitrogen and phosphorus and soil losses. see here some sustainable managements Rodrigo-Comino, J., Terol, E., Mora, G., Giménez-Morera, A., & Cerdà, A. (2020). Vicia sativa Roth. Can Reduce Soil and Water Losses in Recently Planted Vineyards (Vitis vinifera L.). Earth Systems and Environment, 1-16.
Response: The last part mainly discussed the limitation of our study for the optimal design of contour ridge system. The previous findings for the optimal design have been discussed in the section Line 538-550 in the revised manuscript. As suggested, the solutions to reduce the nitrogen and phosphorus have been added in the last part in the revised manuscript(L556-563).

Reviewer 3 Report
Dear authors, I consider that this paper is very interesting and the number of runs, enough to obtain some key conclusions.
The title is too long and does not mention that you used lab experiments. However, in the intro and discussion, I miss some mentions to real cases.
The calibration of the rainfall simulator must be better explained.
Also, photos before and after the experiments must be included.
The conclusions are too long, please, reduce them. See more comments in my attached pdf.

Author Response
1.The title is too long and does not mention that you used lab experiments. However, in the intro and discussion, I miss some mentions to real cases.
Response: We agreed that the title was too long. As suggested, the length of title has been reduced. The title has been changed to “Effect of Ridge Height, Row Grade, and Field Slope on Nutrient Losses in Runoff in Contour Ridge Systems under Seepage with Rainfall Condition”.
In our study, seepage with simulated rainfall experiments were conducted. If simulated rainfall experiment was added in the title, the title became longer. We believed that more important words than simulated experiments, such as nutrient loss, influential factors, and experiment condition, should be mentioned in the title.
We have seriously revised and improved contents in “introduction” and “discussion” section in the revised manuscript. These information for the sustainable development goals, connectivity, roughness have been added in “introduction” section. Importantly, recent information for soil surface roughness, the effect of seepage on flow hydraulic, the effect of row grade on seepage and runoff, the effect of field slope on soil loss, solutions to reduce seepage and soil loss in contour ridge system have been added in “discussion” section in the revised manuscript.
2.The calibration of the rainfall simulator must be better explained.
Response: As suggested, more description for rainfall simulator calibration has been added in the revised manuscript (L193-198).
3.Also, photos before and after the experiments must be included.
Response: As suggested, photos before and after the experiments have been added in the revised manuscript (Figure 3).
4.The conclusions are too long, please, reduce them. See more comments in my attached.
Response: As suggested, the conclusions have been reduced in the revised manuscript (L575-592).
5.the title is too long, please, reduce the length.
Response: As suggested, the title has been reduced. The title has been changed to “Effect of Ridge Height, Row Grade, and Field Slope on Nutrient Losses in Runoff in Contour Ridge Systems under Seepage with Rainfall Condition”.
- Line 13, over the world?
Response: We checked previous researches, and found that few information was available for the characteristics of nutrient loss and the influential factors within contour ridge systems under seepage with rainfall condition over the world.
- Our results demonstrated that...
Response: As suggested, these words “Our results demonstrated that” have been added in the revised manuscript (L20).
- one decimal in the percentages in all the text please.
Response: We believed that two decimals in the percentages were more exact to describe the difference, especially for the difference being not very pronounced.
- Line 27, this is suitable for all the land uses and over the world?
Response: The optimal design is suitable for contour ridge system for controlling nitrogen and phosphorus losses under seepage with rainfall condition over the word, especially for these hilly areas.
- Line 36, over the world?
Response: According to your suggestion, we checked the corresponding reference, and found that in China, nutrient losses of up to 81% for nitrogen and 93% for phosphorus have been discovered in runoff arising from agriculture-related activities, not over the word. To make the description more clearly, we have edited the sentence in the revised manuscript (L41).
- Line 48, I would say "hillslope".
Response: We agreed that seepage flow that occurs on hillslope is generally modeled in laboratory by supplying water from the upper side of experimental plot under a certain water head. To make the description more clearly, the sentence has been edited in the revised manuscript (L53-54).
- Line 66, check these papers where this topic is discussed too... in order to discuss a little the connectivity processes: https://onlinelibrary.wiley.com/doi/full/10.1002/esp.4385; https://link.springer.com/article/10.1007/s12665-019-8505-8
Response: As suggested, we checked these corresponding references for connectivity. Some information for connectivity has been added in the revised manuscript (L71-72, L75-76, L78).
- Line 77, talk about "roughness"\
Response: As suggested, the information for roughness has been added in the revised manuscript (L87-88).
- Line 103, apply where?
Response: The findings in our study not only were reasonable for the study region, but also for other regions at which contour ridge system was the main tillage management and seepage was easy to occur.
- why this? In Table 1
Response: Contour ridge system includes a combination of ridges, adjacent furrows, and the topography of the field where it is implemented. Thus, nutrient loss is affected by geomorphological factors (ridge height, row grade, and field slope) other than the length and steepness of the row sideslope. Our field investigations showed that the range of row grade, field slope and ridge height was 2–10°, 5–15°and 8.0–16.0 cm respectively. The values of the three factors at five code gradients were ±1.682, ±1 and 0 according to the quadratic orthogonal rotation combination design. Firstly, the minimum and maximum values for ridge height (8.0 and 16.0 cm), row grade (2° and 10°), and field slope (5° and 15°) at the corresponding code gradients of 1.682 and -1.682 were determined. Then, the corresponding factor values for code gradients of 1 and 0 were calculated according to the equation(1)
- Line 159, hillslope
Response: We simulated nutrient loss on hillslope within contour ridge system under seepage with rainfall condition. To make the description more clearly, information for hillslope has been added in the revised manuscript (L212-213).
- Line 173, why this?
Response: To avoid the influences of large gravel and plant roots, the used soil was passed through a 10.0-mm sieve. To make the description more clearly, the sentence has been edited in the revised manuscript (L220).
16.Line 225, which software?
Response: Thanks for your suggestion. The software (DPS7.05) has been added in the revised manuscript (L270).
- Line 414, roughness?
Response: The different combination of ridge geometry (ridge height) and contour microtopography (row grade and field slope) results in varied roughness, and then affect nutrient loss. The information for roughness has been added in the revised manuscript (L477-478).

Round 2
Reviewer 3 Report
The authors included a lot of new improvements in this paper. For me, it is ready to be accepted.